**COMMENT**

# Finding the potency in planarians

Bret J. Pearson [ID] [1,2✉]

Planarian flatworms are well-known for their regenerative ability, which is dependent on a large population of adult stem cells, called neoblasts, at least some of which are pluripotent. Here, two recent studies are compared that have begun to address this fundamental question of whether all or only some neoblasts are pluripotent.

Over the past two decades, freshwater planarians have found a resurgence in the laboratory in North America due to their incredible ability to regenerate any body part[1]. The regenerative ability of planarians is dependent on a large population of adult stem cells called neoblasts, which have made planarians an attractive model system for understanding fundamental stem cell biology[2]. In principle, because planarians can regenerate any missing body part from virtually any amputation fragment along the anterior–posterior axis, it has been long-established that neoblasts are collectively totipotent—that is, neoblasts together can remake all missing tissues, including the germline. A key study in 2011 showed that a single neoblast could be totipotent when transplanted into an animal with no stem cells[3]. Since then, there has been a flurry of research investigating how specific cellular lineages are regulated and what heterogeneity, if any, exists within the stem cell population. Recently, two studies have further elucidated the potency of neoblasts, which begins to answer whether totipotent stem cells are a unique subpopulation[4] or whether most/all neoblasts retain high levels of potency and plasticity[5]. Here, I will compare and contrast these studies.

Although the planarian field is relatively small, it has powerful tools and genomic resources at its disposal. Single-cell RNA-sequencing has opened up the field, and multiple, high-quality cellular atlases exist to directly address the molecular heterogeneity in the planarian stem cell population at the RNA level[6,7]. Over the past decade, many studies have shown in situ molecular heterogeneity of neoblasts where the pan-neoblast marker *piwi-1* can be co-expressed with differentiated cellular markers for multiple cell types[8–12]. These data assume that a given *piwi-1+ cellmarker +* neoblast is committed to differentiation into the *cellmarker+* cell type and has withdrawn from the cell cycle. Further functional data have shown that key transcription factors are co-expressed in subsets of *piwi-1 +* neoblasts, and when removed by RNAi, the given cell lineage is lost. This has been demonstrated for the factors: *zfp-1*, which is required for differentiation into the epithelial lineage[9]; *myoD* for the muscle lineage[13]; and *hnf4* for an endodermal lineage[14], for a few examples. From these studies, the working model is a classical top-down cellular hierarchy, which has a pluripotent, clonogenic neoblast (cNeoblast) at the top and progressive restriction through other lineages (Fig. 1). What is unknown from these studies is whether a *piwi-1 + cellmarker +* neoblast is truly committed (irreversibly) to a given lineage or whether it can retain plasticity for multiple lineages (or even pluripotency)? Alternatively, if a totipotent neoblast sits at the apex of the hierarchy and has a distinct transcriptional cell state, then it may be detectable by single-cell sequencing. The two studies here take different approaches to investigate the molecular signature, if any, of pluripotent neoblasts and whether their potency becomes restricted.

[1] Hospital for Sick Children, Program in Developmental and Stem Cell Biology, Toronto, ON M5G10A4, Canada. [2] Present address: School of Medicine, Department of Pediatrics, Papé Family Pediatric Research Institute, Oregon Health & Science University, Portland, OR 97239, USA. ✉email: pearsobr@ohsu.edu

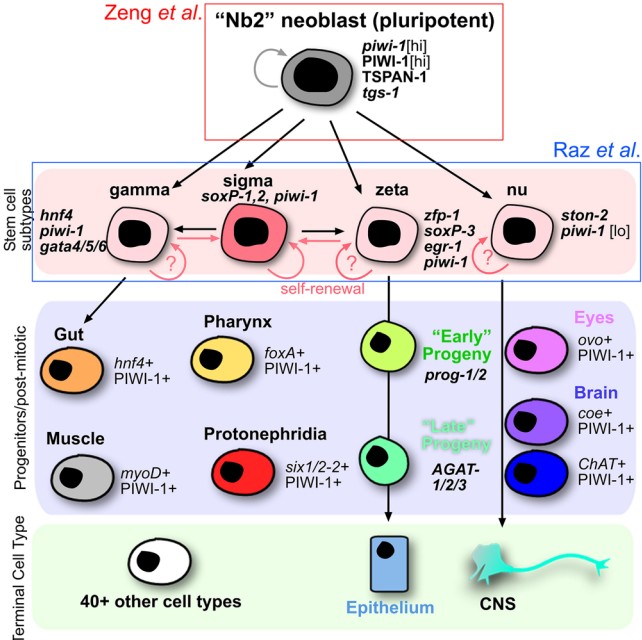

**Fig. 1 Working model of the planarian stem cell hierarchy.** At the top of the stem cell hierarchy in planarians must sit a pluripotent stem cell. Zeng et al. (red box) show support that this cell expresses *tgs-1* and can be prospectively isolated by TSPAN-1 protein expression. Below the pluripotent neoblast may sit various molecular subtypes of neoblasts thus far described in planarians. Relatively little is known about the self-renewal or potency of any cell types at the stem cell subtype level, but the sigma class is likely to contain some pluripotent stem cells and is colored darker than the other cells. Raz et al. (blue box) demonstrate that the cNeoblast may not exist in the hierarchy, and that stem cell subtypes may have the ability to retain pluripotency and switch between stem cell types. Raz et al. also show that as stem cells exit the cell cycle, their commitment to a progenitor state and lineage is likely to occur. Some progenitor cells for specific tissues and lineages have been found and others are not yet associated with a given lineage (no connecting lines).

## Recent findings

In the first study by Zeng et al., the authors specifically isolated and sequenced ~7000 neoblasts with the goal of detecting a gene signature of a pluripotent neoblast, the potency of which was then functionally validated by gold-standard single-cell transplantations[4]. Based on sequencing, the authors detected 12 distinct neoblast subtypes in silico based on the similarity of gene expression. Importantly, the authors detected all previously identified neoblast subtypes. The authors then focused on a large subclass of neoblasts that they could not classify and found that this subtype, called Nb2, expressed high levels of a cell surface protein homolog of *tetraspanin* (*tspan-1*). Importantly, the authors made an antibody to TSPAN-1 and could, for the first time, prospectively isolate a TSPAN-1+ neoblast subpopulation by flow cytometry. To test the potency of TSPAN-1+ neoblasts, single cells were transplanted into hosts devoid of stem cells to test multilineage potential. The authors conclusively demonstrated that some of the TSPAN-1+ stem cells could restore the stem cell compartment, and thus, were functionally pluripotent.

While the Zeng et al., study found a method and gene signature to enrich pluripotent stem cells, there are some other interesting observations. First, while TSPAN-1- cells could not rescue the stem cell compartment, the TSPAN-1+*piwi-1*+ stem cells could only rescue the stem cell compartment of ~25% of animals following transplant. In the 2011 study, the rescue efficiency of single-cell transplants isolated only by morphology and without a

cell surface marker was ~5%, so this study was a marked improvement in enriching for pluripotent neoblasts (or simply enriching for *piwi-1*+ cells). However, it remains unknown whether the 25% rescue in the Zeng study reflects the difficulty of the method (i.e., accidental killing of the transplanted stem cell), or whether this reflects true biological differences in potency. If it accurately reflects biology and only 25% of TSPAN-1+ stem cells are pluripotent, then there is much more room to hone in on the exact pluripotent stem cell population. Second, the Zeng study did not find a molecule that functioned specifically to maintain the TSPAN-1 population. Removal of TSPAN-1 function did not show loss of stem cells in a homeostatic context, and thus it remains unknown whether removal of TSPAN-1+ neoblasts would also remove pluripotency. Finally, the authors found that 89% of TSPAN-1+ cells were also *piwi-1*+, showing a high correlation of TSPAN-1 protein with the stem cell compartment, although the transcript for *tspan-1* itself was difficult to detect at homeostasis. In the end, the Zeng model is attractive because the authors found TSPAN-1+ stem cells distributed throughout the stem cell compartment. Thus, virtually any injury fragment would inherit a pluripotent neoblast to restore any missing cell types (Fig. 2a).

The second study, performed by Raz et al., stratified the >12,000 *piwi-1*+cells previously sequenced into the cell cycle stage based on gene expression[5,6]. Further, they sequenced several thousand new *piwi-1*+ cells taken from the 2C flow cytometry gate (representing G1/G0 stem cells) and the 4C gate (representing G2/M stem cells). The authors then examined the expression of known fate-specifying transcription factors (FSTFs) and observed an increase in FSTF expression as stem cells proceeded through the cell cycle. The authors showed that the 2 cells produced by a division often have an asymmetric expression of an FSTF in the two daughter cells: one that remains *piwi-1*[hi] and FSTF− and the other that is *piwi-1*[low]FSTF+. Through careful analyses, the authors show, surprisingly, that FSTF+ stem cells can give rise to FSTF− stem cells, implying that fate specification may either be reversible or simply adopted by a daughter cell at G2/M and that many or most *piwi-1*[hi] stem cells are pluripotent (Fig. 2b). The Raz model is attractive because pluripotency can be accessed by most stem cells, and thus, these would be present in any given amputation fragment.

Interestingly, Raz et al. find that *tspan-1*+ (assayed using the additional co-expressed transcript *tgs-1*) stem cells largely express FSTFs toward neural fates and are not simply an FSTF−, pluripotent cell state as was suggested by Zeng et al. However, it should be noted that the Raz et al., study was based on RNA expression (and investigating *tgs-1* as a proxy for TSPAN-1). In contrast, Zeng et al. used prospective neoblast isolation based on protein expression investigating TSPAN-1. Thus, while the studies seem at odds, it remains possible that both are correct and that *tgs-1*+ stem cells are a mix of neural-specified and pluripotent. This could also explain the relatively low rescue percentage by a single-TSPAN-1+ neoblast in transplants at 25%.

## Outlook

Despite some key insights into planarian stem cell pluripotency and fate specification, it remains unknown when a reversible specification becomes an irreversible commitment. It also remains unknown what expression of an FSTF in a G1 stem cell means. There is precedent in the literature from other organisms that cell fate transcription factors can turn on in the mother stem cell and the permanent expression is retained only in the differentiated daughter cell. For example, in *Drosophila* embryonic neuroblasts, FSTF expression is often seen in the mother neuroblast, but the expression is only retained in the daughter cell

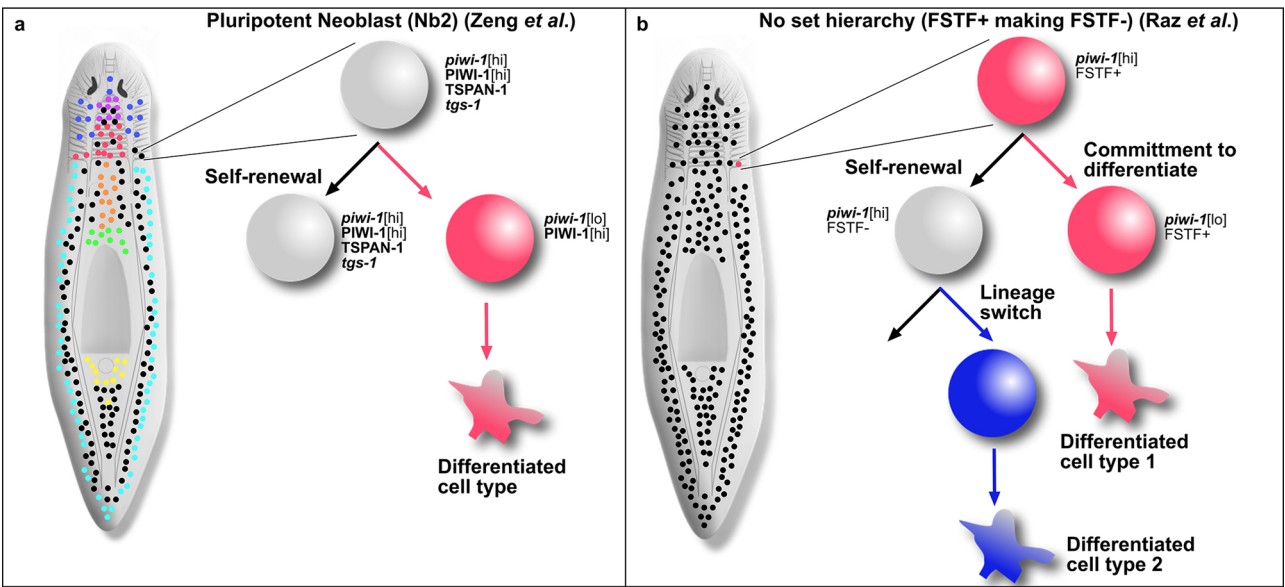

**Fig. 2 Possible interpretations of potency in planarians. a** Zeng et al., suggest that the Nb2 in silico cluster of neoblasts is the pluripotent population in planarians, which can be prospectively purified by TSPAN-1 protein expression and transplanted into hosts devoid of stem cells. Nb2 cells are distributed throughout the body axis (black dots) and specialized subtypes of neoblasts (various colored dots) are made from them in a traditional hierarchy. **b** Raz et al., show that although neoblasts can express factors that make them appear specialized (pink dot; FSTF+), they in fact can give rise to pluripotent-looking neoblasts (black dots; FSTF−), which then make different lineages as well (blue). In this case, a neoblast may appear specialized (pink dot), but it can readily switch back to pluripotent state (black dots). It is a combination of cell cycle stage and down-regulating *piwi-1* transcript the authors propose leads to true lineage commitment.

during an asymmetric division[15]. Similarly, in the vertebrate retina, neural progenitors cycle through different expression states of the cell type they are making, yet will retain potency while the daughter differentiates[16]. So, while Raz et al. conclude that specified stem cells can change FSTF marker expression following a division, this is not unprecedented.

Although both of these studies bring us closer to understanding pluripotency and fate decisions in planarian stem cell lineages, both have limitations that can only be addressed with additional methods to either indelibly mark specific stem cell populations, and/or prospectively isolate them through more cell surface markers. For example, if a *zfp-1+* stem cell could be isolated confidently by a specific cell surface marker, it could be transplanted and a cellular clone analyzed. If it were to give rise to the expected result of only making epithelial clones, this would give high confidence that *zfp-1+* zeta neoblasts are indeed irreversibly committed to the epidermal lineage. Other subtypes of planarian stem cells in Fig. 1 could similarly be tested in this way. The converse result would also be highly informative if *zfp-1+* stem cells could go back and produce cell types of any lineage and rescue animals devoid of stem cells, demonstrating the pluripotency of any stem cell subtype. Unfortunately, these tools do not yet exist and without them, it is unclear how much further single-cell genomics can take us towards answers.

Despite the recent studies, the biggest unknown in planarian stem cell biology remains trying to understand how a single stem cell can give rise to different cell types of multiple lineages. Is this process controlled extrinsically to the stem cell, or do all pluripotent stem cells cycle through making a specific order or ratio of differentiated cell types? To ask this another way, can the pluripotent stem cell sense what differentiated cell needs to be made in a particular location and respond accordingly (i.e., a stem cell near the gut will be biased to make gut), or will it always make a set order of fates (i.e., regardless of location, a stem cell will make a gut cell, then an epidermal cell, then a neuron)? There is

much work to be done to answer these fundamental questions to resolve where potency lies in planarian stem cell lineages.

**Reporting summary**. Further information on research design is available in the Nature Research Reporting Summary linked to this article.

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

## Competing interests

The author declares no competing interests.
