## [Peer Review File · Communications Biology]

Reviewers' comments:

Reviewer #1 (Remarks to the Author):

Planarian stem cells are capable of regenerating whole animals. In 2011, it was demonstrated that a single stem cell was capable of restoring all of the animal's cells, given sufficient time. This finding revealed the existence of pluripotent adult stem cells. A couple of recent papers have attempted to pinpoint the identity of this elusive cell using single-cell RNA sequencing.

This overview describes the findings, remaining unknowns, and unresolved issues in these two papers. Overall the content is sufficient and well written, but there are a few minor suggestions that will improve clarity for readers outside the field (and for those in the field too).

In the second-to-last paragraph (lines 116-123) it seems like answering these questions about the dependency of regeneration actually hinges on being able to confidently identify pluripotent cells through molecular labeling techniques. These two papers use different approaches and yet we are still left with an unsatisfying picture. The author makes a point to call out potential differences in the approaches (lines 97-104), but it might be worth mentioning explicitly that understanding the biology of these pluripotent cells requires their actual identification, which both studies kind of fall short of doing. I think the author implies this by suggesting that an indelible label will help to track the progeny of stem cells, but to me this comes after knowing which cells to label! Beyond the protein vs transcript difference mentioned, the other key difference is that Zeng uses full animal rescue while Raz just analyzes a few cells generating colonies. These experimental differences might also distort the conclusions from their analyses.

It would be helpful to outline more explicitly the limitations for both of these papers. In my opinion, one would expect that if TSPAN really is that essential for pluripotency, then knocking it down might result in regeneration defects or a loss of stem cells, and it doesn't seem to (nor does *tg5-1*). It may be worth mentioning this. On the other hand, the primary shortcoming of the Raz paper is that all of the analysis is done strictly in homeostasis, and it remains completely unknown how or whether these dynamic states would change during regeneration.

Line 34/35: To properly set the stage for the rest of the review, it would help to describe more clearly the very low percent rescue in the 2011 paper, in terms of numbers. This will help to impress the significance of the 25% rescue seen by Zeng to those not familiar with the low efficiency they reported.

Reviewer #2 (Remarks to the Author):

Planarian stem cells (neoblasts) form a very active and accessible stem cell system that can generate any cell type of the planarian body, and thus is a great model to investigate questions surrounding stem cell dynamics and potency. This manuscript is a very nice and timely writeup on recent developments in the field of planarian stem cell research. The author discusses findings of two recent publications that take very different approaches to address the question of pluripotency among the neoblasts, and places the findings in the context of previous research. It also outlines further outstanding questions and directions for the field.

Two things could be addressed to more precisely articulate the intricacies of the field, and both have to do with the dimension of time in the interpretation of pluripotency.

The primary limitation is that the manuscript appears to lean heavily into the idea that there are two types of stem cells that can be separated: pluripotent ones, and lineage-primed ones. It is well beyond discussion that the stem cell population collectively is pluripotent, and also that there are individual

stem cells that are able to generate all cell types and thus are pluripotent. It is also clear that there are stem cells that express features of differentiated cell lineages, and thus can be considered specialized. However whether these are two distinct groups of cells or rather represent cell phases is still not resolved. In fact, one of the two publications (Raz et al) proposes that the undifferentiated pluripotent cells may be widely present in G1, but rare in G2/M. This publication suggests that the undifferentiated state can be regenerated at each cell division by segregating differentiation factors into one of the daughter cells. It did not prove that the daughter with differentiation marks will not be pluripotent, nor that the undifferentiated cell is, but it does clearly suggest that there may be a temporal fluidity to the pluripotency of the neoblasts. It is a bit confusing that the schematic does not include this possibility. To help readers understand the intricacies of these investigations, it would be good to add an alternative model that includes this option, and more clearly sketch out these two possibilities in the text.

The second limitation is that the nature of TSPAN as a neoblast marker is not thoroughly discussed. While it is the only membrane protein marker available for neoblast isolation, and thus is a valuable tool to the field, there are considerable questions about its nature. The tspan transcript is for example not detected in homeostatic animals, and only comes up upon amputation, suggesting that this is not a stable permanent marker of pluripotent cells. In addition, upon amputation tspan goes up in piwi-high cells, but also in piwi-low cells, suggesting that it is not uniquely expressed in the piwi-high cells that are proposed to be pluripotent in this study (Zeng et al). Again, based on the data it appears that TSPAN marks a cell phase rather than a stable cell population.

Overall this review will help readers think more deeply about questions surrounding the identification of pluripotency and should be of interest to planarian researchers and stem cell scientists alike.

Minor:

P2 line 53 This already introduces the pyramid model for stem cell hierarchy, but does not leave much space for an alternative interpretation.

P3 line 79 This quoted ratio (5/232) is for transplantation of the X1(FS) population. The TSPAN-population actually gave no colonies. Of note, the X1(FS) is a quite poor enrichment strategy and only 22% of the cells in this population were even piwi-1+. This may be a significant part of the explanation for why this transplantation strategy had such poor outcome.

P3 line 94 It is not so much that the fate specification is reversible, but rather that it appears to be connected to cell cycle, and thus is (for one of the daughters) cyclic.

P3 lines 97-104 It is unlikely that the discrepancy between the Raz and Zeng studies is due to the difference between analyzing RNA and analyzing protein. This would require the tgs/tspan RNA to remain present in neuronal specified cells while the protein is not translated, and the protein to be present in pluripotent cells where the RNA is absent. The more likely explanation is that tgs and tspan are two different genes, and that only 35% of tspan+ cells expressed tgs-1. Further the Raz study shows no quantification of what fraction of the tgs+ cells express neuronal markers. So indeed, it is very possible that both studies are correct.

P4 line 116 Alternative: I would say that this is another major question - in addition to the distribution of pluripotency through the neoblast population. Further, it has been shown that positional information matters at least to some extent, as for example eye progenitors are not formed in the posterior.

P4 line 125 "depth of the planarian stem cell hierarchy" - It is not clear to me what this means.

P5 The sigma class was defined as a mixed bag and would definitely include the cNeoblast. It may be easier for the schematic to either ignore them or to add the "sigma" label at the cNeoblast cell.

Thanks to the reviewers for the thoughtful comments. In the revised version, the major changes were to address some points of clarity that were raised in both the text and the models. Although the text of the commentary can only be 1500 words, I have attempted to make as many suggested changes as would fit. A point by point explanation is below, and new text in the manuscript is in red.

Reviewer #1 (Remarks to the Author):

Planarian stem cells are capable of regenerating whole animals. In 2011, it was demonstrated that a single stem cell was capable of restoring all of the animal's cells, given sufficient time. This finding revealed the existence of pluripotent adult stem cells. A couple of recent papers have attempted to pinpoint the identity of this elusive cell using single-cell RNA sequencing.

This overview describes the findings, remaining unknowns, and unresolved issues in these two papers. Overall the content is sufficient and well written, but there are a few minor suggestions that will improve clarity for readers outside the field (and for those in the field too).

In the second-to-last paragraph (lines 116-123) it seems like answering these questions about the dependency of regeneration actually hinges on being able to confidently identify pluripotent cells through molecular labeling techniques. These two papers use different approaches and yet we are still left with an unsatisfying picture. The author makes a point to call out potential differences in the approaches (lines 97-104), but it might be worth mentioning explicitly that understanding the biology of these pluripotent cells requires their actual identification, which both studies kind of fall short of doing. I think the author implies this by suggesting that an indelible label will help to track the progeny of stem cells, but to me this comes after knowing which cells to label! Beyond the protein vs transcript difference mentioned, the other key difference is that Zeng uses full animal rescue while Raz just analyzes a few cells generating colonies. These experimental differences might also distort the conclusions from their analyses.

It would be helpful to outline more explicitly the limitations for both of these papers. In my opinion, one would expect that if TSPAN really is that essential for pluripotency, then knocking it down might result in regeneration defects or a loss of stem cells, and it doesn't seem to (nor does tgs-1). It may be worth mentioning this. On the other hand, the primary shortcoming of the Raz paper is that all of the analysis is done strictly in homeostasis, and it remains completely unknown how or whether these dynamic states would change during regeneration.

I agree with the reviewer here and have added a short paragraph on the limitations of the studies (which were fairly cryptic in the previous version). I disagree with trying to guess at what an RNAi phenotype might be for tspan-1 or tgs-1 as there is often a disconnect between useful markers and phenotypes (e.g. piwi-1; particularly in the hypomorphic situation we have with planarian RNAi). It should also be noted that it is a stretch to think the Raz study was strictly in "homeostasis" due to the nature of how irradiation changes gene expression.

Line 34/35: To properly set the stage for the rest of the review, it would help to describe more clearly the very low percent rescue in the 2011 paper, in terms of numbers. This will help to impress the

significance of the 25% rescue seen by Zeng to those not familiar with the low efficiency they reported.

I have added in the numbers (full animal rescue = 7/130 X1(FS) transplants – although that is between sexual and asexual strains, so not totally comparable).

Reviewer #2 (Remarks to the Author):

Planarian stem cells (neoblasts) form a very active and accessible stem cell system that can generate any cell type of the planarian body, and thus is a great model to investigate questions surrounding stem cell dynamics and potency. This manuscript is a very nice and timely writeup on recent developments in the field of planarian stem cell research. The author discusses findings of two recent publications that take very different approaches to address the question of pluripotency among the neoblasts, and places the findings in the context of previous research. It also outlines further outstanding questions and directions for the field.

Two things could be addressed to more precisely articulate the intricacies of the field, and both have to do with the dimension of time in the interpretation of pluripotency.

The primary limitation is that the manuscript appears to lean heavily into the idea that there are two types of stem cells that can be separated: pluripotent ones, and lineage-primed ones. It is well beyond discussion that the stem cell population collectively is pluripotent, and also that there are individual stem cells that are able to generate all cell types and thus are pluripotent. It is also clear that there are stem cells that express features of differentiated cell lineages, and thus can be considered specialized. However whether these are two distinct groups of cells or rather represent cell phases is still not resolved. In fact, one of the two publications (Raz et al) proposes that the undifferentiated pluripotent cells may be widely present in G1, but rare in G2/M. This publication suggests that the undifferentiated state can be regenerated at each cell division by segregating differentiation factors into one of the daughter cells. It did not prove that the daughter with differentiation marks will not be pluripotent, nor that the undifferentiated cell is, but it does clearly suggest that there may be a temporal fluidity to the pluripotency of the neoblasts. It is a bit confusing that the schematic does not include this possibility. To help readers understand the intricacies of these investigations, it would be good to add an alternative model that includes this option, and more clearly sketch out these two possibilities in the text.

I agree that it would be great to cover every possible model, however, due to space constraints and my lack of ability to fit more models into the scenario, I have only been able to make minor changes to the model to try to incorporate “fluidity”.

The second limitation is that the nature of TSPAN as a neoblast marker is not thoroughly discussed. While it is the only membrane protein marker available for neoblast isolation, and thus is a valuable tool to the field, there are considerable questions about its nature. The tspan transcript is for example not detected in homeostatic animals, and only comes up upon amputation, suggesting that this is not a stable permanent marker of pluripotent cells. In addition, upon amputation tspan goes up in piwi-

high cells, but also in piwi-low cells, suggesting that it is not uniquely expressed in the piwi-high cells that are proposed to be pluripotent in this study (Zeng et al). Again, based on the data it appears that TSPAN marks a cell phase rather than a stable cell population.

I don't think this is worth dissecting *tspan-1* expression as 89% of TSPAN+ cells are piwi-1+. Trying to talk about the *tspan-1* transcript or lack of RNAi phenotype isn't germane to the discussion of potency and non-quantitative *in situ*s can never tell you where something isn't expressed.

Overall this review will help readers think more deeply about questions surrounding the identification of pluripotency and should be of interest to planarian researchers and stem cell scientists alike.

Minor:

P2 line 53 This already introduces the pyramid model for stem cell hierarchy, but does not leave much space for an alternative interpretation.

This is true, but a model is just a model and has to be tested. Hierarchical models of stem cell differentiation are the standard in the field and are assumed true until proven otherwise.

P3 line 79 This quoted ratio (5/232) is for transplantation of the X1(FS) population. The TSPAN-population actually gave no colonies. Of note, the X1(FS) is a quite poor enrichment strategy and only 22% of the cells in this population were even piwi-1+. This may be a significant part of the explanation for why this transplantation strategy had such poor outcome.

Thanks for catching this. I have changed the text around the low *piwi-1* expression in the X1(FS) and noted that the TSPAN-1- population did not contain pluripotent neoblasts in their experiments (although a negative result).

P3 line 94 It is not so much that the fate specification is reversible, but rather that it appears to be connected to cell cycle, and thus is (for one of the daughters) cyclic.

P3 lines 97-104 It is unlikely that the discrepancy between the Raz and Zeng studies is due to the difference between analyzing RNA and analyzing protein. This would require the *tgs/tspan* RNA to remain present in neuronal specified cells while the protein is not translated, and the protein to be present in pluripotent cells where the RNA is absent. The more likely explanation is that *tgs* and *tspan* are two different genes, and that only 35% of *tspan*+ cells expressed *tgs-1*. Further the Raz study shows no quantification of what fraction of the *tgs*+ cells express neuronal markers. So indeed, it is very possible that both studies are correct.

I strongly disagree with these statements. Neoblasts are well known to have high levels of RNA binding proteins, such as *bruli-1*, *vasa*, other dead-box helicases, etc., which are known in other systems to hold mRNAs untranslated in granules. In addition, this is a very common stem cell theme such as in mammalian neural stem cells that transcribe many differentiation genes, but they are simply not

translated (i.e. RNAseq of these cells cannot easily distinguish the stem cell population) doi: 10.1016/j.stem.2012.06.010 or any of the papers from Freda Miller's lab on this topic.

As for the analysis between *tspan-1* and *tgs-1* in either study, I agree that they cannot be reconciled between papers at this point.

P4 line116 Alternative: I would say that this is another major question - in addition to the distribution of pluripotency through the neoblast population. Further, it has been shown that positional information matters at least to some extent, as for example eye progenitors are not formed in the posterior.

But by changing positional information without any injury, eye progenitors can be made from other regions *de novo*. I've never seen evidence that AP/DV position has any effect on pluripotency. I agree though that position should have some effects, and have added in a line about performing these same experiments but from regionally-isolated TSPAN-1+ cells.

P4 line 125 "depth of the planarian stem cell hierarchy" - It is not clear to me what this means.

I have removed this paragraph for clarity and space.

P5 The sigma class was defined as a mixed bag and would definitely include the cNeoblast. It may be easier for the schematic to either ignore them or to add the "sigma" label at the cNeoblast cell.

Unfortunately, because sigma cells were never specifically isolated and transplanted, they cannot be considered "cNeoblasts", yet they are molecularly distinct and in the publication record, so they need to be in the hierarchy somewhere. I tried to illustrate their difference with a darker color and known self-renewal ability. I have added this caveat to the figure legend.

Reviewers' comments:

Reviewer #1 (Remarks to the Author):

My comments have been addressed and I only have one minor grammatical suggestion:
In lines 98-100, there are two colons, which is weird construction and can be easily changed.

Reviewer #2 (Remarks to the Author):

The review remains a nice evaluation of two recent and important papers in the planarian stem cell field. I appreciate that the views expressed are those of the author and not necessarily have to match with mine. However maybe my main reservation did not come across well, as it remains in place for this revised version of the manuscript. I will try to phrase it in a different way:

The author evaluates two papers of which one (Zeng) strongly relies on the existence of a hierarchical stem cell system, and the other (Raz) draws the existence of such hierarchy into question. The author clearly acknowledges this distinction. However by framing the review early on in terms of a hierarchical model and drawing figure 1 as a "top-down cellular hierarchy", it appears that the decision is already made to favor one paper over the other. I think it is fine to come to this conclusion after the evaluation, but it seems odd to close the door from the start. It would be cleaner to move the hierarchical model of the stem cell system to Figure 2A, as this all is part of the same hypothesis. Figure 2B could be worked out to more clearly show the meandering stem cell state as opposed to the clear hierarchy of 2A.

Maybe the author did not intend the first figure as a statement of a hierarchical stem cell system, but rather as a flow chart of stem cell states. In that case it would be good to not refer to it as a hierarchical system.

I completely understand the author's response that it isn't possible to discuss every possible model within the constraints of this short review, but in my opinion the review really has to discuss the two contrasting models that are at the core of these two papers.

I hope that this clarifies my concern. If the author is of a different opinion I will respect that though.

Reviewers' comments:

Reviewer #1 (Remarks to the Author):

My comments have been addressed and I only have one minor grammatical suggestion:
In lines 98-100, there are two colons, which is weird construction and can be easily changed.

Fixed, thanks!

Reviewer #2 (Remarks to the Author):

The review remains a nice evaluation of two recent and important papers in the planarian stem cell field. I appreciate that the views expressed are those of the author and not necessarily have to match with mine. However maybe my main reservation did not come across well, as it remains in place for this revised version of the manuscript. I will try to phrase it in a different way:

The author evaluates two papers of which one (Zeng) strongly relies on the existence of a hierarchical stem cell system, and the other (Raz) draws the existence of such hierarchy into question. The author clearly acknowledges this distinction. However by framing the review early on in terms of a hierarchical model and drawing figure 1 as a “top-down cellular hierarchy”, it appears that the decision is already made to favor one paper over the other. I think it is fine to come to this conclusion after the evaluation, but it seems odd to close the door from the start. It would be cleaner to move the hierarchical model of the stem cell system to Figure 2A, as this all is part of the same hypothesis. Figure 2B could be worked out to more clearly show the meandering stem cell state as opposed to the clear hierarchy of 2A. Maybe the author did not intend the first figure as a statement of a hierarchical stem cell system, but rather as a flow chart of stem cell states. In that case it would be good to not refer to it as a hierarchical system.

I completely understand the author’s response that it isn’t possible to discuss every possible model within the constraints of this short review, but in my opinion the review really has to discuss the two contrasting models that are at the core of these two papers.

I hope that this clarifies my concern. If the author is of a different opinion I will respect that though.

I understand your concern, but if you read the accompanying text, it states that this is a model to be tested and that we don’t know where pluripotency lies. The reality is that there is a functional unit called the cNeoblast, that is functionally clonogenic. It absolutely sits at the top of all differentiation hierarchies. The only issue is whether the cNeoblast is a distinct cell state (or even exists), or whether any *piwi-1*[hi] cell has pluripotency. Everything downstream of the stem cell level in Fig. 1 hierarchy is well supported, as is the fact that we never observe dedifferentiation in this system. So, planarian cells form from some sort of cellular hierarchy, and most of that is captured in the figure. I have added to the figure legend to highlight what each study is supporting.